# QTL study reveals candidate genes underlying host resistance in a Red Queen model system

**Maridel Fredericksen** \*, **Peter D. Fields**, **Louis Du Pasquier, Virginie Ricci, Dieter Ebert**

University of Basel, Department of Environmental Sciences, Zoology, Basel, Switzerland

\* maridel.fredericksen@unibas.ch

## Abstract

Specific interactions of host and parasite genotypes can lead to balancing selection, maintaining genetic diversity within populations. In order to understand the drivers of such specific coevolution, it is necessary to identify the molecular underpinnings of these genotypic interactions. Here, we investigate the genetic basis of resistance in the crustacean host, *Daphnia magna*, to attachment and subsequent infection by the bacterial parasite, *Pasteuria ramosa*. We discover a single locus with Mendelian segregation (3:1 ratio) with resistance being dominant, which we call the F locus. We use QTL analysis and fine mapping to localize the F locus to a 28.8-kb region in the host genome, adjacent to a known resistance supergene. We compare the 28.8-kb region in the two QTL parents to identify differences between host genotypes that are resistant versus susceptible to attachment and infection by the parasite. We identify 13 genes in the region, from which we highlight eight biological candidates for the F locus, based on presence/absence polymorphisms and differential gene expression. The top candidates include a fucosyltransferase gene that is only present in one of the two QTL parents, as well as several Cladoceran-specific genes belonging to a large family that is represented in multiple locations of the host genome. Fucosyltransferases have been linked to resistance in previous studies of *Daphnia–Pasteuria* and other host–parasite systems, suggesting that *P. ramosa* spore attachment could be mediated by changes in glycan structures on *D. magna* cuticle proteins. The Cladoceran-specific candidate genes suggest a resistance strategy that relies on gene duplication. Our results add a new locus to a growing genetic model of resistance in the *D. magna–P. ramosa* system. The identified candidate genes will be used in future functional genetic studies, with the ultimate aim to test for cycles of allele frequencies in natural populations.

## Author summary

Identifying the genes under selection is often a necessary step toward understanding the processes that drive evolution. In the case of coevolving hosts and parasites, the host genes under selection are those that confer resistance to the parasite. Here, we aim to identify genes conferring resistance in a coevolving host–parasite system. We map a newly discovered resistance locus to a region adjacent to a previously described resistance supergene, and we validate the locus with additional genetic crosses. By comparing the genes present

**Data Availability Statement:** Scripts for the QTL analysis, RepeatModeler analysis, codon-based sequence alignment, differential expression analysis, and transcript mapping as well as

associated data files are available on GitHub (https://github.com/maf8a/Flocus_QTL) and archived through Zenodo (doi: 10.5281/zenodo.7437819). Assembled and annotated haplotypes of the F-locus region, as well as additional Sanger sequencing data from iF, are available on GenBank (xF accession number: OP831933; iF accession number: OP831934; Sanger accession numbers: OP795832, OP795833, OP795834).

**Funding:** This work was funded by the Swiss National Science Foundation (https://www.snf.ch), grants 310030B_166677 and 310030_188887 to DE. The funders had no role in study design, data collection and analysis, decision to publish, or preparation of the manuscript.

**Competing interests:** The authors have declared that no competing interests exist.

in resistant versus susceptible hosts and by analyzing gene expression data, we identify eight biological candidates. One of the top candidates represents a newly identified gene family that is only found in closely related species and is duplicated in several areas in the genome, and another top candidate strengthens a working hypothesis that resistance might depend on sugar molecules. This work broadens our perspective on the complexity and diversity of resistance loci in this host–parasite system, and it pinpoints intriguing candidates that will be tested in future gene knock-out experiments. Follow-up population genetic studies will help us better understand how parasites coevolve with their hosts in natural populations.

## Introduction

Understanding how diversity is maintained in natural populations is a major goal of evolutionary biology. One way that diversity may be maintained is through coevolution with parasites. For example, the Red Queen hypothesis predicts host and parasite allele frequencies cycle under negative frequency-dependent selection (NFDS), meaning that common genotypes are selected against. A key prerequisite to testing various models of coevolution, including the Red Queen hypothesis, is to identify the loci determining host resistance and parasite infectivity, because these are the loci under selection during coevolution [1]. Despite their importance for understanding coevolution, genetic variants responsible for resistance/susceptibility polymorphisms have only been identified in a few systems, such as R genes in plants [2–4], MHC genes mediating HIV progression in humans [5,6], and viral resistance genes in *Drosophila* [7,8].

The cyclically parthenogenetic freshwater crustacean, *Daphnia magna*, and its bacterial parasite, *Pasteuria ramosa*, have been studied extensively as a natural host–parasite system likely coevolving under NFDS [9–11]. Early studies suggested resistance in this system may have a relatively simple genetic basis. Variation in host resistance depends largely on a single step in the infection process, in which parasite spores attach to a susceptible host's cuticle [12]. This spore attachment is determined by the combination of host and parasite genotypes, with negligible environmental effects [13–15]. Furthermore, genetic crosses demonstrated that host resistance to two *P. ramosa* genotypes shows Mendelian segregation [11]. The combination of Mendelian segregation and little to no effects of the environment suggested that resistance may be controlled by few loci with strong effect.

However, subsequent studies with additional *P. ramosa* genotypes have begun to reveal an increasingly complex genetic architecture of host resistance. The current model includes five resistance loci (A, B, C, D, and E), located on three separate chromosomes and explaining resistance to four *P. ramosa* genotypes (C1, C19, P15, and P20). Moreover, resistance to each genotype is explained by at least two of the loci, which interact through dominance and epistasis [16–18]. An additional layer of complexity is the attachment sites: it was first thought that *P. ramosa* attaches exclusively to the foregut of its host (this is the case for genotypes C1, C19, and P20), but genotype P15 was found to attach to the host's hindgut [18]. A recent study found that *P. ramosa* can infect its host through at least three additional attachment sites [19], suggesting that studies so far have only scratched the surface of the diversity of host–parasite interactions in this system. Despite the insights into the genetic architecture of resistance, mapping efforts have not yet identified the gene(s) responsible for host resistance, in part because the regions containing the loci are structurally complex [16,17].

For this study, we investigate *D. magna* resistance to a fifth *P. ramosa* genotype, P21, which shows a distinct pattern of attachment specificity compared to the four genotypes studied

previously. Genotype P21 offers a new opportunity to map the genetic basis of resistance, and we aim to identify candidate genes that can be tested in functional studies. Initial QTL analyses revealed a single, strong peak on a 2.3-Mb scaffold. Subsequent fine mapping narrowed the QTL to a 28.8-kb region, which we call the F-locus region. The F-locus region contains 13 putative genes (positional candidates) and is adjacent to, but distinguished from, a 50-kb *Pasteuria*-resistance region that was previously described as a supergene (due to apparent lack of recombination), and which contains the A, B, and C loci [16]. By analyzing structural differences between the F-locus haplotypes from the QTL parents, and by comparing gene expression data, we highlight eight of the 13 genes as biological candidates. Our findings reveal candidate genes that can be tested in functional studies, and they also add another locus to our current genetic model, thus bringing us a step closer to understanding the genetic basis of host resistance in this system.

## Results

### Resistance polymorphism maps to a 28.8-kb region downstream of the ABC supergene

The F2 panel of *D. magna* clones had been previously created by crossing clone Xinb3 (reference genotype), susceptible to *P. ramosa* genotype P21, with clone Iinb1, resistant to P21 [20]. In preparation for QTL mapping, we scored the clones in this F2 panel for their hindgut attachment phenotype (attachment positive = susceptible, attachment negative = resistant) to *P. ramosa* genotype P21. The F2 panel consists of a "core panel", comprising a random set of F2 clones created from a genetic cross, and an "extended panel", comprising a specific subset of F2 clones, chosen in a previous study, that are susceptible to *P. ramosa* genotype C19 [16,18]. In total, 178 of 340 (52%) F2 clones were scored as susceptible to P21, including 41 of 191 (21%) in the core panel and 137 of 149 (92%) in the extended panel (S1 File). The 150:41 ratio (79%:21%) of resistant to susceptible clones in the core panel is not significantly different from the 3:1 ratio expected under a classic genetic cross with Mendelian segregation ($\chi^2$ = 1.27, $p$ = 0.26). These results suggest resistance to P21 is dominant, as is also the case for resistance to C19 [21], and resistance to P21 is possibly influenced by only one major-effect locus. Moreover, the very high percentage of susceptible clones in the extended panel suggests that the resistances to *P. ramosa* genotypes C19 and P21 are linked in some form.

Initial interval mapping with a single-locus model revealed a major QTL (Fig 1A) with a peak LOD score of 63.5 (p-value between 0.0 and 0.004 (permutation test), 5% LOD threshold = 3.74) at position 157 centiMorgans (cM) on linkage group 3 in the genetic map (S1 File) created from *D. magna* draft genome version 2.4 (from QTL parent clone Xinb3, GenBank accession: GCA_001632505.1). Setting the major QTL as a covariate suggested the presence of an interacting locus on linkage group 4, but multiple-QTL mapping revealed that the single-QTL model consistently yielded the maximum penalized LOD score (full QTL analysis procedure available on GitHub). This single peak spanned 426 kb between the flanking markers scaffold00288_965 and scaffold01464_354 at positions 154 cM and 161 cM, respectively, and this major QTL explained 57% of the variance in the attachment phenotype (Fig 1A, S1 File). An effect plot revealed that heterozygotes were largely resistant (Fig 1A, inset), reinforcing our suggestion that resistance to P21 is dominant.

We next performed fine mapping by recombination breakpoint analysis, in which we scored additional markers within the QTL region and checked for a perfect association between genotype and phenotype. This approach allowed us to narrow our region of interest from the 426-kb QTL peak to a 28.8-kb region (Fig 1B, S1 and S2 Files). Fine-mapping markers U0_55 and D16 (S1 File) define the closest recombination breakpoints at positions

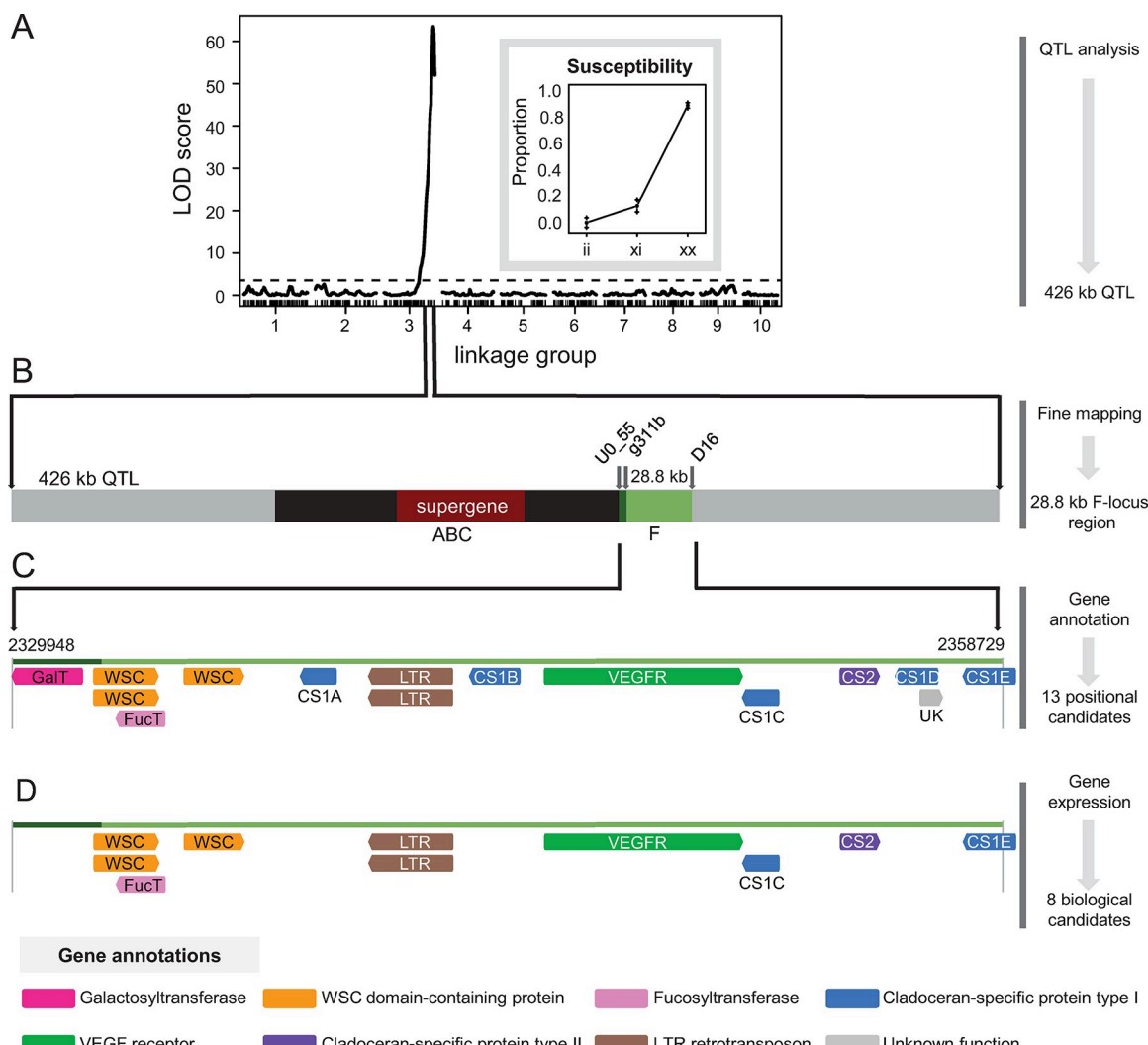

**Fig 1. QTL mapping of the polymorphism in *Daphnia magna* resistance to *Pasteuria ramosa* genotype P21.** Overview of the workflow used in this study to narrow in on candidate genes underlying a resistance polymorphism. Using QTL mapping and recombination breakpoint mapping with additional markers, we associated the polymorphism in resistance phenotype (attachment of *Pasteuria ramosa* genotype P21 to the hindgut of *Daphnia magna* hosts) to a 28.8-kb region at the end of linkage group 3 (*Daphnia magna* genome version 2.4) which we call the F-locus region. **A)** Results of the single-QTL genome scan (i.e., interval mapping), showing LOD score ($\log_{10}$ likelihood ratio) plotted at each SNP marker position in the *D. magna* genetic map. A strong peak with LOD score 63 explains 57% of the phenotypic variance in resistance to P21. The horizontal dashed line indicates the calculated threshold (LOD = 3.74) used to identify significant ($\alpha = 0.05$) QTL. Effect plot (inset) of the SNP marker with the highest LOD score, showing proportion of F2 clones in each genotype group (ii = homozygous for Iinb1 parent clone; xi = heterozygous; xx = homozygous for Xinb3 parent clone) that are susceptible to hindgut attachment by P21. Error bars indicate ± 1 standard error. **B)** Enlarged schematic view of the 426-kb QTL containing the resistance locus (*D. magna* genome version 3.0). The fine-mapped location of the F locus, delimited by SNP markers U0_55 and D16, is shown in green. This F-locus region slightly overlaps with the previously described ABC region (black), which is delimited by SNP marker g311b. The ABC region includes the 50-kb ABC supergene (red), which contains the A, B, and C loci. **C)** Enlarged schematic of the F-locus region, which lies between positions 2,329,948 and 2,358,729 on contig 000011F and contains 13 putative genes (colored by functional annotation), plus two splice variants, in the reference genome (version 3.0) of clone Xinb3. The region colored in dark green indicates the overlap with the ABC region. **D)** Sequence comparisons and differential gene expression analysis between the two QTL parent clones identified eight biological candidate genes in the F-locus region.

2,329,948 and 2,358,729 of contig 000011F in the *D. magna* draft genome version 3.0 (from QTL parent clone Xinb3; BioProject ID: PRJNA624896; Fields et al., in prep.). We call this newly discovered interval the "F-locus region" (i.e., the region containing the F locus), following from the previously described loci A, B, C, D, and E [16–18]. The F-locus region contains

13 annotated protein-coding genes (in Xinb3 genome version 3.0), including those that encode putative glycosyltransferases, cell wall integrity and stress response component (WSC) domain-containing proteins (predicted to bind to sugar molecules [22]), a long terminal repeat (LTR) retrotransposon, a vascular endothelial growth factor (VEGF) receptor, and several uncharacterized genes (Fig 1C). Six of the uncharacterized genes (colored in blue and purple in Fig 1C) were found to be part of a large family that occurs in multiple locations of the *D. magna* genome (S1 Fig and S1 Methods) and seems specific to Cladocera [23], the order comprising waterfleas [24]. The F-locus region is located at the end of its linkage group, directly downstream of (and overlapping by 2,541 bp) the 130-kb ABC region (previously called PR-locus), which contains the 50-kb ABC supergene that underlies resistance to *P. ramosa* genotype C19 [19]. One gene in the F-locus region (a putative galactosyltransferase (GalT); colored in magenta in Fig 1C) occurs in the 2.5-kb region overlapping the ABC region. The ABC supergene is, however, about 17 kb distant from the F-locus region (Fig 2).

There is a small chance that apparent differences in resistance to C19 and P21 occurred because of phenotyping errors, which can never be fully excluded. To validate our fine-mapping data and thus confirm that the F locus is indeed independent of the closely located A, B, and C loci (S1 Methods), we selfed selected F2 clones to produce F3 offspring. We selected two QTL F2 panel clones that were expected to have experienced a recombination event between the ABC supergene and the F locus, resulting in one locus being homozygous and the other heterozygous. We selfed the two selected F2 clones and phenotyped the cloned F3 offspring. If recombination had occurred between the loci, the heterozygous loci should segregate in Mendelian proportions. All offspring of one F2 clone (nr. 693) tested as susceptible to C19, but showed Mendelian segregation of resistance to P21, with 8 susceptible (= 28%) and 21 resistant F3 offspring genotypes (S2 File). These results confirm that the resistance phenotypes to *P. ramosa* genotypes C19 and P21 are coded by two separate loci, located close to each other.

We also noted a slight distortion of segregation for resistance to *P. ramosa* genotype P15 hindgut attachment, which is coded by the D locus; two of the tested F3 offspring of F2 clone 693 were susceptible, even though the F2 parent was homozygous recessive (genotype dd; note that resistance to P15 is recessive), which predicts zero susceptible offspring (see S2 File). Previous work suggested that P15 resistance is mainly affected by the D locus, but partly affected by the dominant allele at the C locus of the ABC supergene [18]. However, in our experiments, the parent clone was homozygous recessive at the C locus (genotype cc), suggesting that the distortion in P15 resistance could instead be caused by the nearby F locus. We could not test this possibility with our current dataset, but we were able to test the opposite interaction: whether a host's genotype at the D locus affects P21 resistance. We therefore tested the QTL panel for evidence of epistasis between the loci coding for resistance to P15 and P21. Results suggest a weak epistatic interaction (LOD = 2.48) in which clones that are homozygous recessive at both D and F loci (genotype ddff) are slightly more likely to be resistant to P21 attachment than clones with a dominant allele at the D locus (genotype D_ff, see S2 Fig).

## The F-locus region shows structural variation between QTL parent clones

Indel polymorphisms and structural rearrangements have resulted in areas of non-homology throughout the F-locus region, including several private genes (unique to either the xF or iF haplotype; see Table 1). Such structural differences represent conspicuous starting points for evaluating potential candidates explaining the resistance phenotype, though it is important to point out that even small genetic differences could be functionally relevant. The most obvious structural difference between iF and xF is that iF is 60% larger (Fig 3). This size difference is due to a combination of duplications and insertions. Both xF and iF have intra-locus

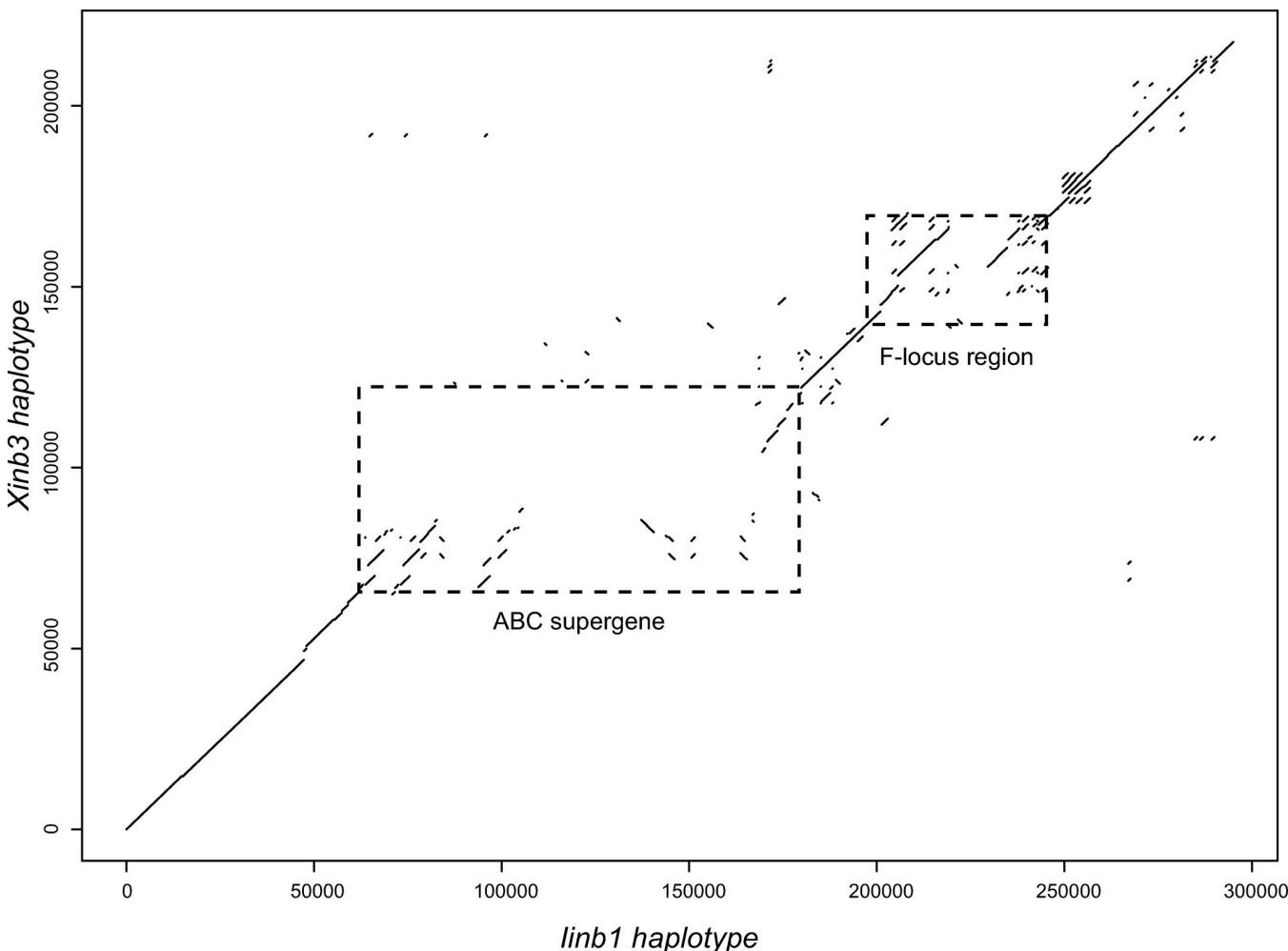

**Fig 2. Compared alignments at ABC supergene and F-locus region.** Dot plots of aligned DNA sequences show regions of homology between QTL parent clones Xinb3 and Iinb1. The depicted alignment includes the ABC region (which contains the ABC supergene) and the neighboring F-locus region, plus 50 kb beyond the F-locus region to aid visual comparison. Axes indicate base-pair position along each respective haplotype. Dashed boxes denote the ABC supergene and F-locus region. This figure has been modified from a previously published figure (Fig 3 in [16]) to specifically compare the location and alignment structure of the ABC supergene and F-locus region.

duplications (i.e., the sequence in question is duplicated within the F-locus region), but these duplications are more abundant in iF compared to xF (Fig 3B and 3C). A 10.5-kb region in xF maps at two positions (10,057–21,095 and 31,603–39,899) in iF (yellow highlight in Fig 3A and 3C). This duplicated region includes homologs to three xF genes (23.9, 23.23, and 23.10). Despite this large duplication, each of the three genes appears to only be expressed once in iF, with the second copy being lost or pseudogenized through gaining or losing stop codons (Fig 3A). Pseudogenization is common following gene duplication, since functional redundancy relaxes selection against deleterious mutations [25–27].

Indeed, all iF private genes appear to be pseudogenes (Table 1), since they were found to be fragments of larger genes located elsewhere in the genome. Some of these fragments (iF 2589, 2599, 2600, and 2601) mapped to genes within the F-locus region. Other fragments (iF 2596 and 2597) mapped to genes outside the F-locus region (i.e., extra-locus duplications, indicated with black and grey bars in Fig 3; see also S3 Fig and S3 File).

**Table 1. Private genes within the F-locus region of the QTL parent clones.**

| Gene (xF) | Gene (iF) | Gene annotation | Complete gene? | Expression (counts) | Protein length | N glycosylation | O glycosylation |
|---|---|---|---|---|---|---|---|
| 23.84-1 | | WSC domain-containing protein 1 | complete | 164.1 | 167 | 0 | 1 |
| 23.20 | | Alpha(1,3) fucosyltransferase C-like | complete | 99.6 | 390 | 6 | 0 |
| 23.94-1 | | LTR retrotransposon | complete | 1.6 | 542 | 5 | 37 |
| 23.113 | | Protein of unknown function | uncertain | — | 67 | 0 | 1 |
| | 2596 | Protein of unknown function | fragment | | | | |
| | 2597 | Protein of unknown function | fragment | | | | |
| | 2598 | Lactosylceramide 4-alpha-galactosyltransferase-like | fragment | | | | |
| | 2599 | VEGF receptor 3 | fragment | | | | |
| | 2600 | VEGF receptor 3 | fragment | | | | |
| | 2601 | VEGF receptor 3 | fragment | | | | |

Summarized characteristics of genes found in either the xF or iF haplotypes of the F-locus region. For genes with splice variants (23.84 and 23.94), we represent the transcript with the longer coding sequence. Several of the annotated genes seem to be incomplete (pseudogenes), because they were found to be fragments of larger genes elsewhere in the genome. Gene expression is presented as mean of normalized counts across all treatments. These values are based on Xinb3 RNA-seq reads mapped to the Xinb3 genome-based transcriptome. Dash indicates that no RNA-seq reads mapped to this gene (xF 23.113).

In addition to annotated genes, we looked for annotated repeat elements in the F-locus region of both the Xinb3 and Iinb1 genomes. Results indicate that xF and iF each contain a private LTR retrotransposon (S4 File and S3 Fig and S3 File). The retrotransposon private to iF is located in a 7.2-kb insertion (position 24,375–31,602 within iF) that did not contain any gene annotations (Fig 3A). The retrotransposon private to xF was annotated as gene 23.94. These retrotransposons contain the most extra-locus duplications in the F-locus region (S3 Fig and S3 File).

Besides the retrotransposon 23.94, xF contains three additional private genes (Table 1). Genes 23.84 and 23.20 are overlapping genes on opposite strands, and they are predicted to encode a WSC-domain-containing protein and a fucosyltransferase, respectively. Overlapping genes share at least one nucleotide in their primary transcripts, and they are widespread in prokaryotic, eukaryotic, and viral genomes [28]. Genes 23.84 and 23.20 were likely lost from iF, as suggested by a segment in iF that maps to the 5' untranslated region of gene 23.84 in xF (orange highlight in Fig 3A). Additionally, gene xF 23.113, which overlaps with gene xF 23.53, is also absent from the annotation of iF (Fig 3A). However, this gene does not show evidence of expression (Table 1), and the predicted protein is quite small (67 aa), thus it may be a pseudogenized gene fragment.

Overall, by comparing the annotated genetic sequences of the QTL parent haplotypes, we identified many structural polymorphisms, including three expressed private genes (xF 23.84, 23.20, and 23.94). These genes may encode functionally relevant differences between the QTL parents, and thus they represent our first three biological candidates.

## Gene sequence and expression comparisons identify additional biological candidates

In order to investigate functional differences between shared genes in xF and iF, we characterized differences in mRNA and protein sequences by calculating similarity measures such as percent identity and piN/piS (nucleotide site diversity within species), and we compared predicted protein structural traits such as number of glycosylation sites (Table 2, S4 and S5 Figs). Most shared genes had very similar sequences, with percent identity above 90% for mRNA

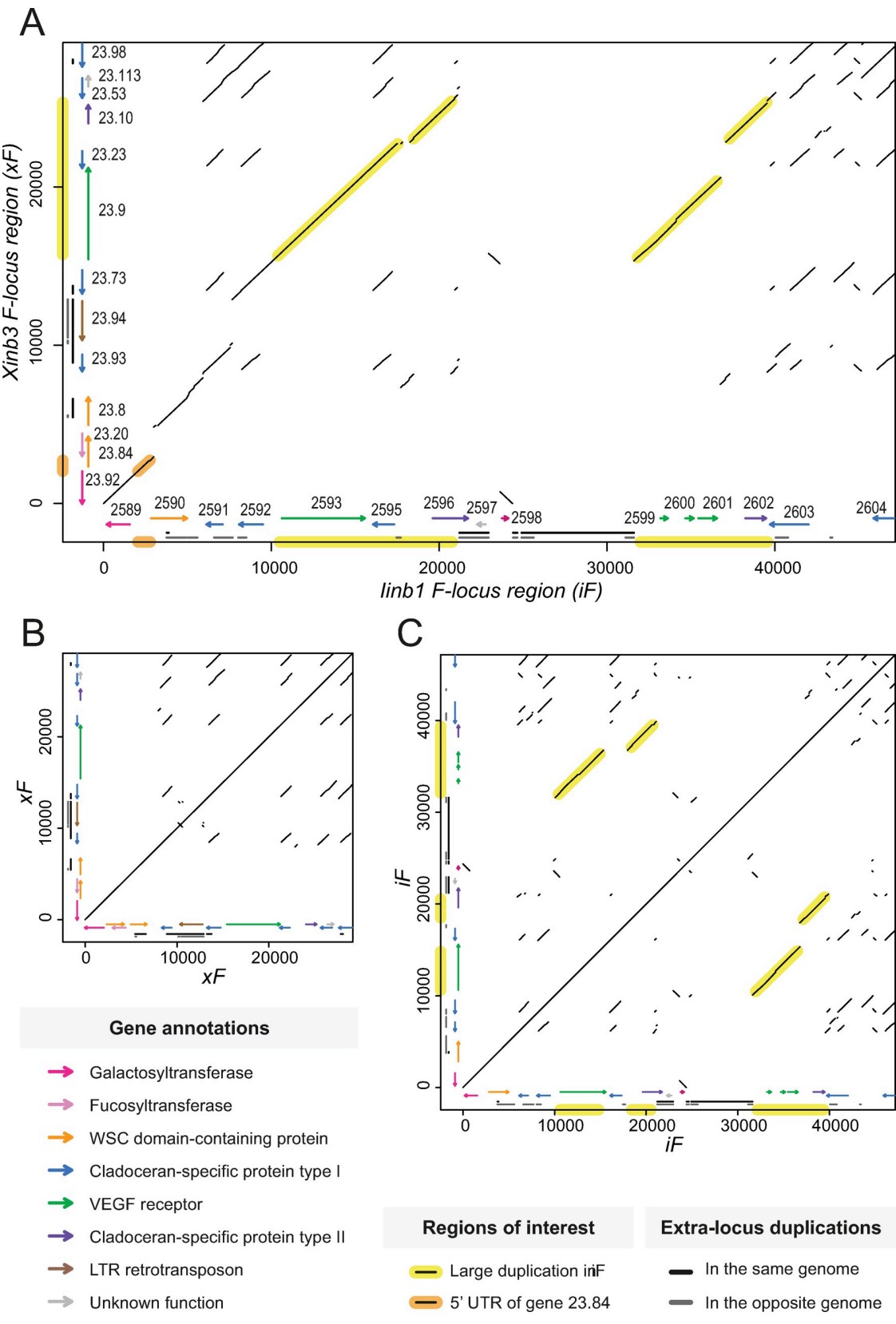

**Fig 3. Dot plots of the F-locus region reveal genetic structure in QTL parent clones.** Dot plots of aligned DNA sequences show regions of homology in the F-locus region from QTL parent clones Xinb3 (xF haplotype, containing susceptible allele) and Iinb1 (iF haplotype, containing resistant allele). Axes indicate base-pair position along each respective haplotype. Homologous regions with a negative slope correspond to inversions. Arrows represent putative genes, colored by functional annotation. Gray and black bars represent regions that are duplicated outside the F-locus region in the same genome and the opposite genome (genome of the other parent clone), respectively. **A)** Alignment of xF and iF. The difference in length of 18.2 kb between xF and iF can be largely attributed to an ~8 kb region of non-homology in iF (~24000–32000) and a large intra-locus duplication in iF (yellow highlight). Part of the 5' untranslated region of gene 23.84 is present in iF (orange highlight), suggesting that this xF private gene was likely lost in iF. **B)** Alignment of xF to itself. Central diagonal line indicates 1:1 homology. Elements outside the central diagonal line indicate intralocus duplications, the largest of which reveal a family of five uncharacterized genes specific to Cladocerans (blue arrows). **C)** Alignment of iF to itself. The abundance of elements compared to panel B indicates that intralocus duplications are more prevalent in iF compared to xF.

and protein (Table 2). Gene xF 23.53 (iF 2603) is an exception, but this gene is not expressed in either parent and we do not consider it further. Nearly all genes had piN/piS values below 1, the one exception being gene xF 23.10 (iF 2602).

The F-locus region contains six Cladoceran-specific uncharacterized genes (xF 23.93, 23.73, 23.23, 23.53, 23.98, and 23.10, and their iF positional homologs; colored in blue and purple in Fig 3A and Table 2) that can be divided into two types belonging to the same large gene family. Protein products from the first five of these genes (CS1) share high sequence similarity (> 40% similarity to at least one other member of this group and > 35% similarity to all members) and likely arose from recent duplication events (S4 and S5 Figs). Topological similarities between these five paralogs include a predicted transmembrane domain near the C-terminal end as well as a shared intracellular amino acid motif YXXC (YXVC and FXVC in our dataset, see S4 Fig), which is thought to be involved in the JAK/STAT signaling pathway [29]. Predicted protein products from genes xF 23.10 and iF homolog, 2602, do not have this STAT motif and have lower sequence similarity to the CS1 paralogs (< 30% similarity to all CS1 proteins; S5 Fig), thus we categorize these as type II Cladoceran-specific proteins (CS2). Despite the similarities in sequence and overall topology between the CS1 paralogs (within each genome), pairs of xF and iF homologs (between genomes) from all six Cladoceran-specific proteins

**Table 2. Shared genes within the F-locus region of the QTL parent clones.**

| | Gene (xF) | Gene (iF) | Gene annotation | Expression (counts) | | % ID mRNA | % ID protein | Protein length | | piN/piS | N glycosylation | | O glycosylation | |
|---|---|---|---|---|---|---|---|---|---|---|---|---|---|---|
| | | | | xF | iF | xF vs iF | xF vs iF | xF | iF | xF vs iF | xF | iF | xF | iF |
| | 23.92 | 2589 | Lactosylceramide 4-alpha-galactosyltransferase-like | 62.0 | 67.2 | 95.18 | 98.90 | 365 | 365 | 0.55 | 4 | 4 | **3** | 2 |
| | 23.8 | 2590 | WSC domain-containing protein 1 | **224.9** | **175.0** | 93.02 | 94.32 | 357 | 352 | 0.37 | 3 | 3 | 4 | 4 |
| | 23.93 | 2591* | Cladoceran-specific protein type 1 (CS1A) | 1.5 | 1.4 | 97.40 | 94.12 | 272 | 272 | 0.83 | 1 | 1 | **9** | 2 |
| | 23.73 | 2592* | Cladoceran-specific protein type 1 (CS1B) | 14.8 | 13.1 | 95.14 | 96.85 | 316 | 316 | 0.29 | **1** | **0** | 11 | 10 |
| | 23.9 | 2593* | VEGF receptor 3 | **144.8** | **163.2** | 83.21 | 96.41 | 946 | 925 | 0.21 | **9** | **12** | 13 | 11 |
| | 23.23 | 2595 | Cladoceran-specific protein type 1 (CS1C) | **6.4** | **0.2** | 86.30 | 93.72 | 239 | 252 | 0.44 | 1 | 1 | **0** | **9** |
| | 23.10 | 2602 | Cladoceran-specific protein type 2 (CS2) | 51.0 | **110.7** | 96.67 | 92.33 | 287 | 287 | **1.73** | 0 | 0 | **0** | **5** |
| | 23.53 | 2603 | Cladoceran-specific protein type 1 (CS1D) | — | — | **54.06** | **45.16** | **298** | **365** | 0.75 | 1 | 1 | 6 | **11** |
| | 23.98 | 2604 | Cladoceran-specific protein type 1 (CS1E) | **12.7** | **5.2** | 97.27 | 96.18 | 314 | 312 | 0.32 | 1 | 1 | **9** | **4** |

Summarized characteristics of genes found in both the xF and iF haplotypes of the F-locus region. Bold type and gray shadings indicate notable differences between xF and iF, with light gray indicating a relatively low value and dark gray indicating a relatively high value. Expression is presented as mean of normalized counts across all treatments. These values are based on Xinb3 and Iinb1 RNA-seq reads mapped to the Xinb3 genome-based transcriptome. Dashes indicate that no RNA-seq reads mapped to this gene. Differential expression is based on Benjamini-Hochberg-adjusted p-value < 0.05 between xF and iF homologs across all treatment groups (see main text and Fig 3 for further information). Percent identities were calculated in relation to length of the aligned region (including gaps). piN/piS values represent nucleotide site diversity within species.

* Genes whose mRNA and protein sequences have been corrected, based on Sanger sequencing data (See S9 Fig).

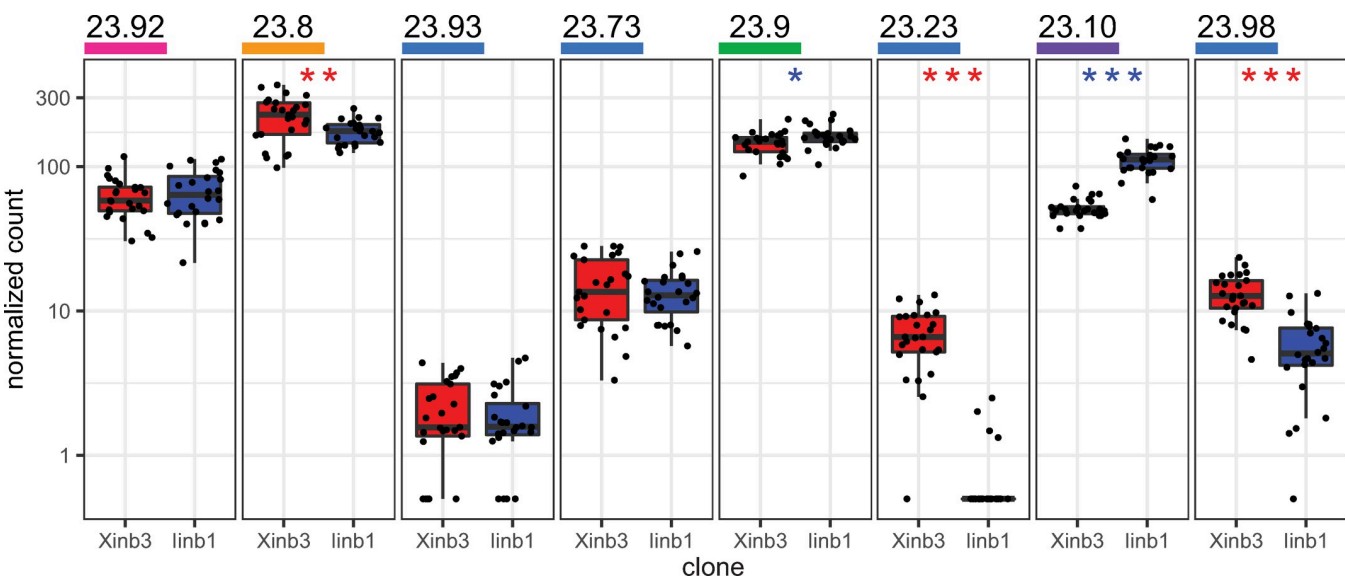

**Fig 4. Differential gene expression of shared genes between QTL parent clones.** Boxplots showing gene expression data collected for a previous study [30], with raw RNA-seq reads from QTL parent clones Xinb3 and Iinb1 mapped to the genome-based transcriptome from clone Xinb3. Gene names are underlined with colors corresponding to functional annotation (see Fig 1). Y-axes from each plot show normalized read counts with a pseudocount of 0.5 added to allow for log-scale plotting. Each plotted point represents a *Daphnia magna* individual from the respective clone. Box edges indicate first and third quartiles, central line indicates median, and whiskers extend to 1.5 x interquartile range. Asterisks indicate genes that show significant differential expression (non-zero logarithmic fold change in mean expression across all treatments combined) between QTL parent clones Xinb3 (n = 25) and Iinb1(n = 24), with clone Xinb3 as reference after correcting for multiple tests: Benjamini-Hochberg-adjusted p-value < 0.05 (*), < 0.01 (**), < 0.001 (***). Asterisks are colored according to the parent clone which shows higher expression (red = susceptible parent Xinb3; blue = resistant parent Iinb1). Wald test statistics and Benjamini-Hochberg-adjusted p-values are as follows: **23.92:** z = 0.425, p = 0.498; **23.8:** z = -2.78, p = 9.96E-03; **23.93:** z = -0.488, p = 0.690; **23.73:** z = -0.898, p = 0.442; **23.9** z = 2.59, p = 0.0167; **23.23:** z = -8.89, p = 4.00E-18; **23.10** z = 13.23, p = 8.51e-39; **23.98:** z = -6.23, p = 1.80E-09. For the same expression data presented separately for each treatment condition, see S6 Fig.

differ in their predicted number of O-glycosylation sites (Table 2), and any of these differences could be functionally relevant for the resistance polymorphism.

In addition to the gene sequence comparisons, we performed a differential gene expression analysis to investigate functional differences between homologs in xF and iF. We used RNA-seq data that had been previously collected from Xinb3 and Iinb1 individuals that were exposed to a variety of stressful conditions [30]. We mapped the raw RNA-seq reads to the genome-based transcriptomes from both clones. We present here the results from the analysis in which all reads were mapped to the Xinb3 transcriptome (Figs 4 and S6, and S5 File). Mapping the reads to the Iinb1 transcriptome produced qualitatively similar results (S7 Fig and S5 File). Because the attachment phenotype depends on the host genotype, rather than on environmental factors [14], we included all treatments in our differential expression analysis so that we could analyze constitutive expression differences (independent of environmental treatment) between resistant and susceptible clones. Such constitutive expression differences were previously shown for genes in the ABC region [16].

Five genes in the F-locus region showed significant differential expression between the QTL parent clones (p-value < 0.05, Fig 4, Table 2 and S5 File). The VEGF receptor gene (xF 23.9) and the CS2 gene (xF 23.10) showed higher expression in the resistant parent (Iinb1). Two CS1 genes, CS1C (xF 23.23) and CS1E (xF 23.98) showed higher expression in the susceptible parent (Xinb3), as did gene xF 23.8, which encodes a putative WSC domain-containing protein. The two genes showing the strongest upregulation in each respective parent clone were the CS2 gene (xF 23.10) in the resistant parent, and gene CS1C (xF 23.23) in the susceptible parent.

In summary, from the 13 positional candidate genes in the F-locus region, we identified eight biological candidates that appear to harbor functional differences and thus may underlie the resistance polymorphism studied here. These eight candidates include three private genes (xF genes 23.84, 23.20, and 23.94) that are present and expressed in only one QTL parent (Table 1). The other five candidates (xF genes 23.8, 23.9, 23.23, 23.10, and 23.98) are shared between both QTL parents (Table 2) but show difference in gene expression (Fig 4).

## Discussion

In this study, we identified and described a single genetic locus, which underlies *Daphnia magna* resistance to *Pasteuria ramosa* genotype P21. The "F locus" shows Mendelian segregation and maps to a 28.8-kb region containing 13 genes. Detailed comparison of gene annotations and expression data from QTL parent clones elevated eight of these positional candidates to biological candidate genes. Putative protein products of these biological candidates include a fucosyltransferase, two WSC-domain-containing proteins, an LTR retrotransposon, a VEGF receptor, and several uncharacterized but related proteins that are unique to the waterflea clade, Cladocera.

### A dominant resistance locus with Mendelian segregation

Our discovery of the F locus adds to the genetic model of *D. magna* resistance to *P. ramosa*. With the newly added F locus, the model now describes the dominance patterns and epistatic interactions of six loci in *D. magna* that explain resistance to five genotypes of *P. ramosa*. Each locus added to this model creates a more complete picture of the genetic variation in *D. magna* resistance. The F locus in particular revealed that hindgut attachment is mediated by different host loci (D or F) depending on the parasite genotype (P15 or P21, respectively). All other known resistance loci mediate attachment to the host's foregut. Additionally, the proximity between the F locus and ABC supergene demonstrated that resistance loci may be located physically close to each other even if they mediate resistance to different attachment sites. Clustering of immune-related genes has been reported for other systems as well, e.g., the MHC genes in vertebrates [31,32] and R genes in plants [33,34], and such clustering may be adaptive, as recombination rates are strongly reduced.

Characterizing the diversity of loci that underlie resistance in this system will help us better understand how these resistance loci arise and are maintained within populations during coevolution. For example, selection on the ABC supergene may influence the F locus and vice-versa because they are physically linked in the genome. Indeed, the flanking sequences of the ABC region (including the F-locus region) were previously shown to have elevated genetic diversity and signals of balancing selection compared to other genomic regions [35]. Future studies may test to what extent the F locus itself contributes to this elevated genetic diversity and signals of balancing selection and to what extent it is influenced by the nearby supergene.

Our analysis showed that resistance at the F locus is dominant when tested against *Pasteuria ramosa* genotype P21. Dominant resistance was also shown for loci B and C against *P. ramosa* genotypes C19 and C1, respectively [16]. In contrast, resistance is recessive for loci D and E against genotypes P15 and P20 [17,18]. For the A locus, on the other hand, dominance depends on the parasite genotype [11]). Although this context-dependent dominance makes the A locus unique regarding resistance to the five parasite genotypes described so far, such a genetic architecture may also be the case for the other resistance loci. Once a greater diversity of *P. ramosa* genotypes [19,36] is incorporated into studies of host resistance, we may find that dominance at other resistance loci also changes with the parasite genotype tested.

The described resistance loci A–E interact with each other epistatically, and the F locus also shows evidence of such interactions. Previous work demonstrated epistasis between loci A, B, C, and E, all of which determine parasite attachment to the host's foregut. The D and F loci, on the other hand, determine attachment to the host's hindgut, and here we found evidence suggesting the F-locus genotype may interact with the D locus (S2 Fig and S2 File). A previous study suggested that the C locus, or another tightly linked locus, may interact with the D locus [18]. Instead, here we suggest this effect may have been caused by the F locus, which is strongly physically linked to the C locus.

Our finding that resistance at the F locus shows Mendelian segregation suggests that the polymorphism has a simple genetic basis. Indeed, each attachment phenotype studied so far is a binary trait mediated by few loci, with negligible environmental effects [16–18]. Thus, if one considers resistance to each parasite genotype as a separate trait, then the underlying genetics may be straightforward. Such an approach may be necessary for uncovering the genetic mechanisms underlying resistance. However, to understand coevolution between *D. magna* and *P. ramosa* in natural populations, the relevant trait to consider is overall resistance, since this is the trait under selection. Resistance to each *P. ramosa* genotype studied so far is coded by different loci, with epistatic interactions between loci. So far, the more we investigate overall resistance to *P. ramosa*, the more complex this trait becomes. While it is not yet clear what this increasing complexity implies for coevolution, it is clear that to understand *Daphnia–Pasteuria* coevolution, we must continue to characterize the diversity of resistance phenotypes.

## Candidate genes to be tested in functional studies

The complexity of *D. magna* resistance to *P. ramosa* does not end with the genetic model of overall resistance; rather, the individual genetic regions containing the resistance loci are also highly complex [16,17], a trend which has hindered efforts to identify candidate genes underlying resistance. In the current study, we were fortunate to find that the F-locus region is largely homologous, which allowed us to compare gene content and gene expression between the QTL parent clones to ultimately identify eight biological candidate genes that may be tested in forthcoming functional studies. These candidates include three private genes of the Xinb3 parent (Table 1) and five differentially expressed shared genes (Table 2).

Given that resistance to P21 hindgut attachment is dominant, can we speculate which of our biological candidate genes is more likely to contain the F locus? Most genes are expected to show higher expression in individuals with the dominant phenotype [37]. These genes are haplosufficient, i.e., one working copy is sufficient for gene function. Genes 23.9 and 23.10 would thus be good candidates to confer resistance, because they show higher expression in the resistant parent, Iinb1, than in the susceptible parent, Xinb3 (Table 2, Fig 2). Alternatively, the gene responsible for the resistance polymorphism could be haploinsufficient. In this case, two copies of the gene would confer susceptibility, and we would expect higher expression in the susceptible parent. Several of our biological candidates would fit these criteria: the three genes private to the Xinb3 (genes 23.84, 23.20, and 23.94), and the three genes that showed higher expression in Xinb3 than in Iinb1 (genes 23.8, 23.23, and 23.10). In contrast to haplosufficiency, haploinsufficiency is relatively rare in nature, with only a few hundred such genes described in humans [38] and a mere 3% of the genes in yeast [39]. Even so, prominent examples of haploinsufficient genes exist in several disease systems. Such examples include the HIV-1 coreceptor CCR5 in humans [40], as well as the A1S subunit of a voltage-gated calcium channel in mice, which affects arenavirus entry and infection [41]. In the following, we use haplosufficiency as one criterion to roughly rank the eight biological candidates. Other criteria include differences in gene expression or protein structure, as well as whether similar genes were previously identified in other *Pasteuria* resistance loci.

Among the eight biological candidates, the gene encoding a type II Cladoceran-specific protein (CS2: gene xF 23.10/iF 2602) stands out for several reasons. First, it was one of only two candidates (the other being the VEGF receptor xF 23.9/ iF 2593) that showed higher expression in the resistant parent (Fig 4), which is the expected differential expression pattern for haplosufficient genes given that resistance is dominant at the F locus. Compared to the VEGF receptor gene (xF 23.9/ iF 2593), the CS2 gene (xF 23.10/ iF 2602) showed larger expression differences between the two QTL parents (Fig 4). Second, this gene is unique among the candidates shared between xF and iF in that it has a strongly elevated piN/piS of 1.79, the only gene in the F-locus region exceeding 1 for this metric (Table 2). Such a high piN/piS value indicates that the xF and iF versions of this gene are remarkably diverged at nonsynonymous sites and suggests that selection might be driving the divergence between these two alleles. Third, the protein sequences differ in their predicted O-glycosylation sites (Table 2). Differential glycosylation has been associated with resistance polymorphisms in other systems (reviewed in [42]). Moreover, differential glycosylation has been linked to the attachment specificity of *P. ramosa*: a predicted N-glycosylation site in the *P. ramosa* gene PCL7 showed perfect association with the parasite's ability to attach to its host [36]. Finally, the CS2 gene belongs to an uncharacterized Cladoceran-specific gene family that has been found in several other locations in the *D. magna* genome, including within the ABC region and the D-locus region [18]. The presence of the Cladoceran-specific gene family in the vicinity of multiple known *Pasteuria*-resistance regions suggests that the evolution of *P. ramosa* resistance in this system may be influenced by gene duplication.

Another strong candidate is the alpha(1,3)fucosyltransferase, coded by xF private gene 23.20. This candidate is of particular interest, because it supports previous findings from the ABC region, in which several putative fucosyltransferase genes were present only in the susceptible QTL parent haplotype [16]. This previous finding led to a working hypothesis that fucosyltransferase genes are among the top candidates for explaining the resistance polymorphism, and our current findings are consistent with this hypothesis. Fucosyltransferases are a family of glycosyltransferases, which are enzymes that assemble glycan chains (polysaccharides) by transferring sugars to another sugar, a protein, or a lipid [43]. Alpha(1,3)fucosyltransferases (including the xF private gene 23.20), in particular, transfer a fucose sugar as the final step in building glycan chains [44–46]. Fucosyltransferases often play a critical role in resistance polymorphisms since they attach sugars mostly at peripheral positions in glycan sequences such that these sugars may interact directly with pathogens attaching to them [47,48]. This is the case in the bacterial pathogen *Helicobacter pylori*, which infects mammalian epithelial cells by attaching to glycans on ABO blood group antigens [49–51].

Thus, one plausible mechanism underlying attachment in the *D. magna*–*P. ramosa* system is that parasite spores may attach to glycosylated proteins on the host cuticle, and variations in glycosylation may confer the variations in attachment phenotype we observe among different host genotypes. Specifically, in the case of attachment by *P. ramosa* genotype P21 to the host's hindgut, we can speculate that the parasite spores may only attach to glycans that contain a fucose transferred by the fucosyltransferase encoded by gene xF 23.20. The presence of fucosyltransferase genes in multiple resistance loci and the plausible molecular mechanism of attachment make gene 23.20 a strong biological candidate. However, note that since the gene is only found in the susceptible parent haplotype xF (and sequence evidence indicates it was likely lost in iF rather than gained in xF), it would need to be haploinsufficient to confer dominant resistance.

Two other candidates (CS1C (xF 23.23/iF 2595) and CS1E (xF 23.98/ iF 2604)) show strong differential gene expression and predicted O-linked glycosylation (Table 2, Fig 4), though both genes are more highly expressed in the susceptible parent, meaning that, like the

fucosyltransferase candidate, these genes would need to be haploinsufficient (or experienced gain-of-function mutations [52]). Like gene CS2 (xF 23.10/ iF 2602), these candidates also encode Cladoceran-specific proteins. Specifically, they belong to the group of five genes that were identified as paralogs of each other, due, in part, to a conserved transmembrane domain followed by a STAT motif (S4 Fig). The STAT motif is implicated in the recruitment and activation of the transcription regulator STAT3 [29], which is part of the JAK/STAT signaling pathway that acts in embryonic development but also in arthropod innate immunity [53,54]. The presence of this STAT motif suggests that the five paralogs in the F-locus region could be part of a signaling pathway; these proteins might communicate between the outside and inside of the cell. It seems unlikely that membrane proteins such as these would interact directly with attachment of *P. ramosa* spores, because the attachment occurs at the surface of the host cuticle, which is non-cellular. However, these proteins could still play a role in modulating host resistance, for example by affecting cuticle organization [55].

The gene candidate encoding a putative vascular endothelial growth factor (VEGF) receptor (xF 23.9/iF 2593) shows the expected pattern of higher gene expression in the resistant parent (happlosufficient), but the expression difference is not very strong. Like the Cladoceran-specific proteins described above, VEGF receptors have a transmembrane domain, giving them access to both the interior and exterior of the cell [56]. VEGF receptors in *Drosophila* play a role in forming the epithelium [57], suggesting that these receptors could also affect development of the cuticle, which is secreted by epithelial cells. Perhaps also in *D. magna*, VEGF receptors like gene 23.9 affect how this host's cuticle develops and thus affect the ability of *P. ramosa* spores to attach to the cuticle. The final three candidate genes (xF 23.94, 23.84, and 23.8) would be haploinsufficient, and they do not stand out among the biological candidates in terms of predicted protein structure or function.

While the above considerations help us rank the eight candidate genes to some degree, none of the given arguments is waterproof. Thus, the next step requires experimental manipulation using genetic tools to knock out the candidate genes and assess the effect on the attachment phenotype [58,59]. Knockout strategies would have to differ depending on the predicted haplosufficiency of the candidate gene: for a haplosufficient candidate like gene CS2 (xF 23.10/ iF 2602), one could select a (resistant) heterozygote and knock out the gene to test whether susceptibility is induced. For a haploinsufficient candidate like the fucosyltransferase gene 23.20, one would choose a (susceptible) homozygote and knock out one copy to test whether resistance is induced. In addition, lectin staining along with glycoproteomics could determine which glycans may be involved in attachment.

## Conclusion

Here we describe the sixth polymorphic resistance locus of *D. magna* against the virulent and wide-spread bacterial parasite *Pasteuria ramosa*. The high resolution of sequence polymorphism we reached for the F locus is new for this system and adds nicely to what we know about the system as a whole. The six loci are believed to be under selection during host–parasite coevolution, and their identification can therefore aid us to shed light on the coevolutionary process. With more of these loci being described, we see more patterns, which may allow us to formulate testable hypotheses. All of the resistance loci show Mendelian segregation when their effects are isolated from other loci, but segregation patterns of all loci can be strongly distorted by epistatic interactions with other loci. Resistance can be dominant or recessive and, in all cases, seems to be unaffected by environmental variation. However, so far, GWAS and QTL approaches were in no case able to pinpoint the actual genetic polymorphism. This is, in part, because the resistance regions are highly diverse, including structural

polymorphisms. Furthermore, resistance loci seem to be placed in regions of low genetic recombination: four of the six loci (A, B, C, and F) cluster tightly together, and the ABC and the E loci seem to be located in supergenes. This reduced recombination hinders our attempts to fine map the location of these resistance loci. High genetic diversity, perfect dominance and epistasis, strong phenotypic effects without environmental effects, and reduced recombination rates have all been suggested to promote Red Queen dynamics, strengthening the suggestion that these loci evolve under negative frequency-dependent coevolution.

Besides the population genetic aspects, with more loci being identified, we can also speculate about the molecular mechanisms at work. Two of the six resistance loci mediate attachment to the hindgut; the other four mediate attachment to the foregut. Both these sections of the gut are of ectodermal origin, meaning they are part of the molted cuticle. Therefore, candidate genes may code either for proteins expressed on the surface of the cuticle or for enzymes that modify these proteins, e.g. by glycosylation. Across the genomic regions containing the six described *D. magna* resistance loci, both these types of proteins are represented, including the fucosyltransferase candidates described here for the F locus. If resistance involves polymorphisms in proteins of both types (presence/ absence of cuticle proteins with or without modification), these protein interactions may give rise to the changing patterns of dominance and epistasis that we observe among the resistance loci. The right combination may determine if the parasite can attach or not.

## Methods

### QTL and fine mapping

Quantitative Trait Locus (QTL) mapping was performed using a standing *Daphnia magna* QTL panel [20]. This panel was created by crossing a Finnish mother clone (FI-Xinb3 (here called Xinb3), susceptible to *Pasteuria ramosa* genotype P21) with a German father clone (DE-Iinb1 (here called Iinb1), resistant to *P. ramosa* genotype P21), and all offspring were genotyped at 1,324 SNP markers (S1 File) based on the *D. magna* draft genome version 2.4 (GenBank accession: GCA_001632505.1). Note that genome version 2.4 was used for QTL mapping only; all fine mapping and subsequent analyses were performed using the more recent version, 3.0 (both genome versions are from clone Xinb3). For the QTL analysis, we tested the hindgut attachment phenotype by spores from *P. ramosa* genotype P21 for a total of 340 clones from the F2 "core" panel (randomly selected F2 lines) and the "extended" panel (F2 clones susceptible to *P. ramosa* genotype C19) [13] using the *Pasteuria* attachment test [14]. An infection experiment confirmed that attachment by P21 spores to the host's hindgut is highly correlated with infection outcome (S8 Fig and S1 File and S1 Methods). The spores from *P. ramosa* genotype P21 were obtained from a *D. magna* population in the Aegelsee, Switzerland [17]. We performed the QTL analysis, which tests the association of the phenotype data to each marker in the genetic map, using the R Statistical Software (version 3.4.3) [60] package qtl (version 1.42–8) [61], with a binary phenotype model and Haley-Knott regression [62] as described previously [21]. As part of the R/qtl analysis, a genome-wide significant ($\alpha$ = 0.05) LOD score was calculated at 3.74 using 1,000 permutation tests. We performed the QTL mapping using both a single-locus model (function scanone) and two-locus model (function scantwo), and the results from these analyses informed our final multiple-QTL mapping (function stepwiseqtl), which employs a penalized LOD score approach for model selection.

In order to fine map the resulting major QTL, we designed additional genetic markers (S1 File) using the Xinb3 reference genome (version 3.0) and the Iinb1 draft genome, and we placed these new markers between the QTL-flanking markers. We then genotyped the F2 clones that had a recombination event within this region (i.e., F2 clones whose genotype differs

between the two markers flanking the major QTL). We analyzed size-polymorphic markers and SNP markers using PCR followed by capillary electrophoresis (Applied Biosystems 3130xl Genetic Analyzer) and Sanger sequencing (Microsynth; Basel, Switzerland), respectively. We then used these genotype data to locate the recombination breakpoints in each of the selected F2 clones, assuming a perfect association between genotype and phenotype.

In order to validate that resistance to *P. ramosa* genotype P21 is conferred by a locus that is separate from the previously characterized ABC supergene [16], we produced F3 offspring (S1 Methods) by selfing several F2 clones that had a recombination event between the C locus (underlying resistance to foregut attachment by *P. ramosa* genotype C19) and the putative F locus (underlying resistance to hindgut attachment by *P. ramosa* genotype P21). We then phenotyped the F3 offspring for resistance to C19 foregut attachment and P21 hindgut attachment to infer the parent clone's genotype according to the segregation pattern of the offspring. Offspring which segregated for P21 resistance but not for C19 resistance were inferred to come from a parent clone that is heterozygous at the F locus and homozygous at the C locus, indicating that the F locus and C locus are indeed distinct. We also tested the F3 offspring for resistance to P15 hindgut attachment in order to check for possible epistasis between the F and D loci. Previous work suggested that loci C and D may interact such that P15 hindgut resistance (mainly coded by the D locus) may be partially explained by the dominant allele at the C locus [18]. In the current experiment, both of the tested F2 clones were homozygous at the C and D loci but potentially heterozygous at the putative F locus. Thus, if the F3 offspring segregate in their resistance to P15, this could due to an effect from the F locus, rather than from the nearby C locus.

## Characterizing the F locus

We characterized differences in the F-locus region between the Xinb3 and Iinb1 haplotypes, here called "xF" and "iF", respectively. We extracted the xF and iF haplotype sequences from the respective genomes, and we aligned these nucleotide sequences to themselves and to each other using LASTZ (version 1.04.03) [63] under default settings. Because much of the nearby ABC supergene consists of duplications [16], we also checked the F-locus haplotypes for intra-locus and extra-locus duplications in both genomes. We define an intra-locus duplication as a sequence that is fully contained in the F-locus region in at least two locations. An extra-locus duplication occurs outside the F-locus region. Specifically, we used BLASTN (version 2.7.1+, blast.ncbi.nlm.nih.gov) to query each F-locus haplotype (xF and iF) against each entire genome. Hits with an e-value of less than $10^{-20}$ were considered homologous [64] and thus classified as a duplication. Besides the BLAST approach, we also used the program RepeatModeler2 (version 2.0.1) [65] with RepeatMasker (version 4.1.0) [66] to identify repeat elements such as transposable elements in both xF and iF haplotypes.

Finally, we annotated the genes in the F-locus region of both xF and iF to help identify biological candidates underlying the resistance polymorphism. Annotations of the QTL parent genomes Xinb3 (version 3.0) and Iinb1 were performed using the MAKER2 pipeline (version 2.31.10) [67], including ab-initio gene predictors by AUGUSTUS (version 3.3.3) [68], SNAP (version 2006-07-28) [69], and GeneMark (version 4.28) [70]. Upon visual inspection of these gene annotations, we determined that several annotated genes were actually fragments of complete genes that occurred elsewhere in the genome, and other annotations were incomplete (e.g., missing UTRs). We manually curated the incomplete annotations using a combination of transcriptomic data, protein structure predictions, and alignment to homologous genes (see S1 Methods and S6 File). Additionally, we corrected the coding sequences of three genes (2591, 2592, and 2593) from clone Iinb1 based on Sanger sequencing data (S9 Fig). All downstream analyses were based on the curated gene annotations.

## Gene and protein sequence comparisons

We compared curated gene and protein sequences between xF and iF haplotypes. We aligned mRNA and protein sequences of positional homologs (determined by relative position in the respective haplotypes) using MAFFT (version 7.450) [71] under the following settings in Geneious (version 2022.1.1) (algorithm: FFT-NS-1, scoring matrix: 200PAM/k = 2 (for mRNA; for proteins, scoring matrix = BLOSUM62), gap open penalty: 1.53, offset value: 0.123) and then calculated percent identity in relation to the length of the aligned region (including gaps) for each gene pair.

Alignments of orthologous coding sequences between xF and iF were made using a custom R script which initially utilized the R (version 4.0.3) package seqinr (version 3.6–1) [72] to import individual coding sequences for each respective genotype, followed by identification of the correct reading frame, and finally, a codon-based alignment using R package prank (version 170427) [73,74]. We calculated piN/piS in DnaSP (version 6.12.03) [75] on each alignment. We also characterized protein topology by predicting presence and location of transmembrane domains using CCTOP (version s.1.00) [76], and by locating putative N- and O-linked glycosylation sites with NetNGlyc1.0 [77] and NetOGlyc4.0 [78], respectively.

## Gene expression analysis

We quantified gene expression for complete genes in the F-locus region from each parent clone using raw RNA-seq reads collected previously for *D. magna* individuals from parent clones Xinb3 (n = 25 individuals) and Iinb1 (n = 24 individuals) [30]. These RNA-seq reads were mapped separately to the Xinb3 and Iinb1 *in-silico* derived transcriptome, i.e., transcript sequences were generated using the aforementioned annotation by using the program gffread (version 0.11.4) [79] to extract transcript sequences from the gff file. Transcripts were mapped and quantified using the program kallisto (version 0.46.1) [80], and counts were analyzed for differential gene expression between clones Xinb3 and Iinb1 using the R/Bioconductor package DESeq2 (version 1.28.1) [81]. P-values were adjusted for multiple testing using the Benjamini-Hochberg method [82].

Scripts for the QTL analysis, RepeatModeler analysis, codon-based sequence alignment, differential expression analysis, and transcript mapping as well as associated data files are available on GitHub (https://github.com/maf8a/Flocus_QTL) and archived through Zenodo (doi: 10.5281/zenodo.7437819). Assembled and annotated haplotypes of the F-locus region, as well as additional Sanger sequencing data from iF, are available on GenBank (xF accession number: OP831933; iF accession number: OP831934; Sanger accession numbers: OP795832, OP795833, OP795834).

## Supporting information

**S1 Methods. Supplementary materials and methods with supporting literature.**
(DOCX)

**S1 Fig. Alignment of Cladoceran-specific protein sequences.** Clustal Omega alignment of amino acid sequences from 53 Cladoceran-specific genes located on contigs 000011F (containing the ABC supergene and the F locus) and 000018F (containing the D locus). cov: coverage; pid: percent identity. Residues of note: cysteine: yellow; charged amino acids: red; serine and threonine: light blue. Consensus sequence is indicated below the alignment and is particularly notable between positions 400 and 720.
(HTML)

**S2 Fig. Effect plot and epistasis test between P15 and P21 resistance.** Effect plot showing mean susceptibility (± 1 *SE*) as a function of genotype at two putative QTL explaining host variation in hindgut attachment of *Pasteuria ramosa* genotype P21. The x axis shows variation at the QTL detected on linkage group 3 (lg3) at position 157.0 (near F locus), and different colors represent the genotypes at the QTL on lg9 at position 105.3 (near D locus). The Xinb3 QTL parent is known to be susceptible to both parasite genotypes, with genotype ff at lg3 and DD at lg9. The Iinb1 QTL parent is known to be resistant to both parasite genotypes, with genotype FF at lg3 and dd at lg9. Table shows estimated support of each QTL after dropping one term at a time from a binary multiple-QTL model with both QTL (lg3 @ 157.0 and lg9 @ 105.3) and an interaction term (lg3 x lg9) between them. For each model term, we give the degrees of freedom (df), the $\log_{10}$ likelihood ratio (LOD) comparing the full model to reduced models, the estimated percent of phenotypic variance explained by the term, and a p-value that is based on the LOD score and assumes a $\chi^2$ distribution of LOD x (2ln10). Note that p-values are pointwise, meaning they do not account for the search over the whole genome. We therefore consider them with caution.
(PDF)

**S3 Fig. Extra-locus duplication mappings.** Extra-locus duplications from the F-locus region and their mapping locations in the respective QTL parent genome (for full BLAST results see S2 File). Duplicated segments (black) are separated and labeled according to the contig to which they map (leading zeroes in contig names are omitted). Colored arrows indicate annotated genes in the F-locus region (see Fig 1 for color code). **A**) Extra-locus duplications from xF and their mapping locations in the Xinb3 (susceptible QTL parent) genome. **B**) Extra-locus duplications from iF and their mapping locations in the Iinb1 (resistant QTL parent) genome.
(PDF)

**S4 Fig. Alignment of five Cladoceran-specific paralogs.** Predicted sequences of five Cladoceran-specific (type I) paralogs from each of xF and iF were aligned using MAFFT in Geneious. The predicted transmembrane domain and STAT motif are indicated (pink and green, respectively).
(PDF)

**S5 Fig. Heatmaps of percent identity for six Cladoceran-specific genes.** Heatmaps comparing percent identities of six Cladoceran-specific genes, including five type I (blue) and one type II (purple) from each of xF and iF. Percent identities were calculated from pairwise alignments of full mRNA sequences (A) and predicted protein sequences (B). Bold black outline indicates positional homologs (compared in Table 2).
(PDF)

**S6 Fig. Differential gene expression of shared genes between QTL parent clones across treatment conditions (mapped to Xinb3).** Boxplots showing gene expression data collected across multiple stressful conditions for a previous study [30], with raw RNA-seq reads from QTL parent clones Xinb3 and Iinb1 mapped to the genome-based transcriptome from clone Xinb3. Gene names are underlined with colors corresponding to functional annotation (see Fig 1). Y-axes from each plot show normalized read counts with a pseudocount of 0.5 added to allow for log-scale plotting. Each plotted point represents a *Daphnia magna* individual from the respective clone. Box edges indicate first and third quartiles, central line indicates median, and whiskers extend to 1.5 x interquartile range. Asterisks indicate genes that show significant differential expression (non-zero logarithmic fold change in mean expression across all treatments combined) between QTL parent clones Xinb3 (n = 25) and Iinb1(n = 24), with clone Xinb3 as reference after correcting for multiple tests: Benjamini-Hochberg-adjusted p-

value < 0.05 (*), < 0.01 (**), < 0.001 (***). Asterisks are colored according to the parent clone which shows higher expression (red = susceptible parent Xinb3; blue = resistant parent Iinb1). Wald test statistics and Benjamini-Hochberg-adjusted p-values are as follows: **A)** z = 0.425, p = 0.498; **B)** z = -2.78, p = 9.96E-03; **C)** z = -0.488, p = 0.690; **D)** z = -0.898, p = 0.442; **E)** z = 2.59, p = 0.0167; **F)** z = -8.89, p = 4.00E-18; **G)** z = 13.23, p = 8.51e-39; **H)** z = -6.23, p = 1.80E-09.
(PDF)

**S7 Fig. Differential gene expression of shared genes between QTL parent clones across treatment conditions (mapped to Iinb1).** Boxplots showing gene expression data collected across multiple stressful conditions for a previous study [30], with raw RNA-seq reads from QTL parent clones Xinb3 and Iinb1 mapped to the genome-based transcriptome from clone Iinb1. Gene names are underlined with colors corresponding to functional annotation (see Fig 1). Y-axes from each plot show normalized read counts with a pseudocount of 0.5 added to allow for log-scale plotting. Each plotted point represents a *Daphnia magna* individual from the respective clone. Box edges indicate first and third quartiles, central line indicates median, and whiskers extend to 1.5 x interquartile range. Asterisks indicate genes that show significant differential expression (non-zero logarithmic fold change in mean expression across all treatments combined) between QTL parent clones Xinb3 (n = 25) and Iinb1(n = 24), with clone Iinb1 as reference after correcting for multiple tests: Benjamini-Hochberg-adjusted p-value < 0.05 (*), < 0.01 (**), < 0.001 (***). Asterisks are colored according to the parent clone which shows higher expression (red = susceptible parent Xinb3; blue = resistant parent Iinb1). Wald test statistics and Benjamini-Hochberg-adjusted p-values are as follows: **A)** z = -1.44, p = 0.204; **B)** z = 2.36, p = 0.0312; **C)** z = -1.11, p = 0.339; **D)** z = 1.24, p = 0.279; **E)** z = -4.61, p = 1.19E-05; **F)** z = 8.51, p = 1.26E-16; **G)** z = -18.82, p = 2.29e-77; **H)** z = 4.80, p = 5.01E-06.
(PDF)

**S8 Fig. Correlation between attachment phenotype and infection outcome.** Scatterplot showing percent of individuals with positive attachment to the hindgut and percent of exposed individuals showing disease symptoms five weeks after infection. Each point represents one of 40 *Daphnia magna* clones from the QTL F2 panel, tested against spores from the *Pasteuria ramosa* genotype P21. The linear regression line is shown in black, with gray shading indicating the 95% confidence interval. Positive attachment was strongly correlated with subsequent infection (Spearman's rho = 0.76, P < 0.001, *n* = 40).
(PDF)

**S9 Fig. Corrections of three Iinb1 gene sequences based on Sanger sequencing data.** Nucleotide sequence alignments showing the iF haplotype extracted from the Iinb1 reference genome (upper sequence in each panel) aligned to Sanger-sequencing data (lower sequence in each panel), which were used to correct the sequences of iF genes 2591 (A), 2592 (B), and 2593 (C) for comparison with xF homologs (summarized in Table 2). Sanger sequencing results include associated chromatograms, colored by the nucleotide base called at the given peak. Annotations above the aligned sequences indicate the consensus sequence (colored by base identity and numbered according to the position in the iF haplotype) and the identity score of the alignments (green indicates 100% identity and gaps indicate polymorphisms). Raw sequencing data can be accessed at the GenBank accession numbers provided at the end of the manuscript. Figure was created using Geneious Prime software.
(PDF)

**S1 File. QTL and Fine-mapping data.  README)** Column descriptions for following tables QTL phenotypes) Attachment test results (*Pasteuria ramosa* P21 hindgut attachment) for all

*Daphnia magna* clones from the QTL panel. Red and blue shading correspond to susceptible (attachment positive) and resistant (attachment negative) phenotypes, respectively. **QTL SNP map**) QTL mapping data containing clone names, attachment phenotype, (1 = susceptible, 0 = resistant), and genetic markers with corresponding genotypes (AA = matching resistant parent, BB = matching susceptible parent, AB = heterozygous). **Infection experiment**) Attachment and infection results for 40 *Daphnia magna* clones from the QTL panel. Thirty individuals from each clone were exposed to spores from *Pasteuria ramosa* genotype P21 and assessed for disease symptoms after five weeks. **Marker primers**) Primer information for all size-polymorphic (microsatellite) and SNP markers used for fine mapping. **Finemap microsat**) Fine-mapping results from size-polymorphic (microsatellite) markers. A subset of clones from the QTL panel was selected for genotyping at eight markers within the QTL region (See S1 Methods for additional description of fine-mapping procedures). Red cells (B) indicate a haplotype matching the susceptible parent Xinb3, dark blue cells (A) indicate a match to the resistant parent (Iinb1), and light blue cells (H) indicate heterozygotes. Clones 177 and 693 (delimited) identify the limits of this fine-mapping effort at markers D42 and U1, respectively. **Finemap Sanger**) Fine-mapping results from SNP markers. Clones 177 and 693 were selected for further fine mapping at 13 additional markers. Recombination breakpoints were localized within markers U0_55 and D16_9.
(XLSX)

**S2 File. Selfing.** We selfed two F2 panel clones (94 and 693) with a recombination event in the F-locus region to test the independence of the ABC supergene and the F locus, and to attempt additional fine mapping. **README**) Column descriptions for following tables. **Collection**) Selfing is performed by collecting ephippia (sexually produced resting eggs) and hatching them after a period of dormancy. Ephippia were collected across four dates in 2019 and 2020. **Attachment tests**) Selfed offspring were tested for resistance to attachment by *Pasteuria ramosa* genotypes C19, P15, and P21. Resistance to these genotypes is known to segregate in the QTL panel. Red and blue shading correspond to susceptible (attachment positive) and resistant (attachment negative) phenotypes, respectively. **Summary**) Summary table of resistance to each of the three *P. ramosa* genotypes. We inferred the genotype of each locus predicted to confer resistance to the parasite genotypes, according to the genetic model of resistance and assuming Mendelian segregation.
(XLSX)

**S3 File. Duplications.** BLASTn output of F-locus haplotypes against the whole genome from each QTL parent. Tables include both intra-locus and extra-locus duplications, and output is sorted by query start position. **README**) Column descriptions for following tables. **xFvsXinb3**) BLASTn output for Xinb3 F-locus region against whole Xinb3 genome (version 3.0). **iFvsIinb1**) BLASTn output for Iinb1 F-locus region against whole Iinb1 genome. **xFvsIinb1**) BLASTn output for Xinb3 F-locus region against whole Iinb1 genome. **iFvsXinb3**) BLASTn output for Iinb1 F-locus region against whole Xinb3 genome (version 3.0).
(XLSX)

**S4 File. RepeatMasker output.  README**) Column descriptions for following tables. **Xinb3**) RepeatMasker output for Xinb3 F-locus region. **Iinb1**) RepeatMasker output for Iinb1 F-locus region
(XLSX)

**S5 File. DESeq2 output (differential gene expression). README**) Column descriptions for following tables. **Mapped to Xinb3**) Output from differential gene expression analysis in which all RNA-seq reads were mapped to the Xinb3 genome (version 3.0). Full results have

been filtered to only include the genes in the F-locus region. **Mapped to Iinb1)** Output from differential gene expression analysis in which all RNA-seq reads were mapped to the Iinb1 genome. Full results have been filtered to only include the genes in the F-locus region. (XLSX)

**S6 File. Hand-curation of gene annotations from F-locus region. README)** Column descriptions for following tables. **Xinb3)** Hand-curation for Xinb3 F-locus region. **Iinb1)** Hand-curation for Iinb1 F-locus region (XLSX)

## Acknowledgments

The authors would like to thank Jürgen Hottinger, Michelle Krebs, Urs Stiefel, and Kristina Müller for assistance in the laboratory, as well as Meret Halter, Pascal Angst, Joana Santos, and Jeremias Brand for assistance with coding. We also thank members of the Ebert group for helpful feedback on the manuscript.

## Author Contributions

**Conceptualization:** Maridel Fredericksen, Dieter Ebert.

**Data curation:** Peter D. Fields, Louis Du Pasquier.

**Formal analysis:** Maridel Fredericksen, Peter D. Fields, Louis Du Pasquier, Virginie Ricci.

**Funding acquisition:** Dieter Ebert.

**Investigation:** Maridel Fredericksen.

**Software:** Peter D. Fields, Virginie Ricci.

**Supervision:** Dieter Ebert.

**Visualization:** Maridel Fredericksen, Virginie Ricci.

**Writing – original draft:** Maridel Fredericksen, Dieter Ebert.

**Writing – review & editing:** Maridel Fredericksen, Peter D. Fields, Louis Du Pasquier, Virginie Ricci, Dieter Ebert.

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
