## [Decision Letter · Decision Letter 0]

2 Aug 2022

Dear Dr Fredericksen,

Thank you very much for submitting your Research Article entitled 'QTL study reveals candidate genes underlying host resistance in a Red Queen model system' to PLOS Genetics.

The manuscript was fully evaluated at the editorial level and by independent peer reviewers. The reviewers appreciated the attention to an important problem, but raised some substantial concerns about the current manuscript. As you will see, this is the unusual situation, where the primary criticisms are about putting the current work in proper context in light of your and others' work. In many respects, much of the present study seems to be framed as the abstract and introduction for a broad review, rather than the focus being on the identification of a very interesting locus, which is the culmination of your efforts. As complimentary as the reviewers are of the work, they are also very critical about the presentation of the manuscript, which in current form does not do a good enough job to highlight the significance of the present study. Indeed, one of the reviewers felt that, in current form, the study may not rise to the level of significance we desire for PLOS Genetics. I do feel that many of these criticisms can be address by a serious rreworking of the framing of the paper. I hope you will do so guided the comments. Based on the reviews, we will not be able to accept this version of the manuscript, but we would be willing to review a much-revised version. We cannot, of course, promise publication at that time.

If you decide to revise the manuscript for further consideration at PLOS Genetics, please aim to resubmit within the next 60 days, unless it will take extra time to address the concerns of the reviewers, in which case we would appreciate an expected resubmission date by email to plosgenetics@plos.org.

[LINK]

We are sorry that we cannot be more positive about your manuscript at this stage. Please do not hesitate to contact us if you have any concerns or questions.

Yours sincerely,

Harmit S. Malik

Academic Editor

PLOS Genetics

Kirsten Bomblies

Section Editor

PLOS Genetics

Reviewer's Responses to Questions

**Comments to the Authors:**

Reviewer #1: The genetic basis of host-pathogen coevolution is a topic of considerable interest. The system studied here is an important model of coevolution, thanks in a large part to the efforts of this research group. Over recent years they have made progress in understanding this system at a genetic level. The significance of this work is increased by the context of these species being a well-characterised model in the field, and this manuscript adds to this literature. It reports a QTL affecting susceptibility to a bacterial parasite in Daphnia. The QTL contains 13 genes, but the authors are unable to identify which of these confers resistance. The QTL mapping panel, genomes and RNAseq data have been published before, frequently in similar studies using different parasite genotypes. The major new data presented here is therefore the phenotyping.

The main clear result of the paper is that resistance to a new pathogen genotype is controlled by a major-effect Mendelian locus that is different from that controlling resistance to previously studied pathogen genotypes. Furthermore, this locus has numerous rearrangements and gene gains/losses, similar to nearby locus controlling resistance to a different genotype.

The efforts to characterise what the gene might be are weaker, as the genes remain firmly candidates. A lot of text is devoted to describing these genes and their expression, but this remains circumstantial. I realise that genetics in this species is challenging, but polymorphisms controlling pathogen resistance have been identified to the gene or nucleotide level in other species. Furthermore, a number of similar QTL studies have been published in this system.

In this context, it is unfortunate the authors do not acknowledge the literature on genetic polymorphisms affecting host resistance. For example, the introduction states “the exact identity of the genetic variants responsible for ,resistance/susceptibility polymorphisms segregating in natural populations remains mostly unknown” [and the rest of this paragraph]. The abstract “molecular underpinnings of the genotypic interactions are largely unknown”. It is poor scholarship to make these statements without citing systems where the statements do not apply, and it detracts from the advances this work might bring to an important model system. There are many cases where “exact identity of the genetic variant responsible for ,resistance/susceptibility polymorphisms segregating in natural populations” is known. Furthermore, in a number of cases these loci confer resistance only to specific pathogen genotypes. The most prominent cases are perhaps plant R genes. In vertebrates, MHC genes are widely studied, down to the structural differences underpinning the genotype-specific HIV-MHC epitope binding that affects progression to AIDS. There are other examples in Drosophila.

Minor comments:

Line 97. It would help to be precise about what defines this 50kB region as a supergene? (inversion breakpoints in genome assemblies, LD, lab crosses?)

Line 139. Citation to justify 1.8 LOD drop (I believe a Broman group paper)

Line 143. The results and methods need to make clear what exactly was done in the ‘fine mapping’. Is this just scoring more markers? If so, how is the confidence interval on the QTL location defined? In the proceeding section a 1.8 LOD drop is used, but there is no mention of the statistics here. I take it this is simply checking for a perfect genotype-phonotype association? It is unclear why this change in statistical approach has occurred – if this is sufficient here, then why use a LOD drop to analyse a near-identical dataset with fewer markers in the earlier paragraph? I don’t doubt the results here, but the analysis being conducted and its justification needs clarifying.

Figure 2 appears to have been published before by these authors in this journal (slightly different parameters, a new dashed box). Is that correct? Maybe there is a good reason to reproduce the result here with minor changes, but not to acknowledge and cite its earlier incarnation seems surprising. Or maybe I am missing something here?

Line 192-249. This could be made considerably more concise – it is hard to see the wood for the tree.

Figure 4. Why not just report the ‘all’ results in the main text? There are lots of apparently irrelevant treatments, making this plot rather complex and hard to read. The sudden appearance of a plot about fish predation is rather befuddling!

Line 313-325. This belongs in the Discussion.

The Discussion spends much time repeating results in this and other papers. It could be made more concise, focussing on broader context to increase impact. The manuscript also begins by stating the importance of this work to our understanding of the process of coevolution. Is there anything to say on this? I understand this to be one of the few species where the matching alleles model applies – might these results suggest this results from tightly linked genes following the gene-for-gene system model?

Three levels of subheadings does not add clarity.

Embedding legends and figures in the text (as opposed to 3 locations) at initial submission is helpful to a reviewer with a small laptop screen…

Reviewer #2: In this manuscript the authors use the Daphnia-Pasteuria system to identify and map an additional genomic region affecting bacterial infectivity in the system. The authors identify eight candidate genes within the region with the potential to explain the differences in bacterial spore attachment between the parents used for QTL mapping. Overall, this work is incredibly thorough and adds an important piece to characterizing the basis for host resistance to Pasteuria infection. Further, the natural context and ecological dynamics within the system make this a very promising system to phenotypically and genotypically characterize coevolutionary dynamics in the wild.

Overall, I think this is an important study within the field and for the system. However, I believe this manuscript struggles somewhat to stand on its own. I fully agree that this system has great potential to characterize the Red Queen in action. But, this study does not directly investigate coevolution. Rather, the focus of this work is mapping the F locus and identifying candidate genes. While I generally agree with everything the authors state about the system relative to the Red Queen, I think the paper could be re-framed somewhat to highlight the work that is presented rather than the potential of the system as a whole. It is currently unclear to me how the work presented here connects to the Red Queen, other than adding an additional locus to track. Further, it is not clear what adding an additional locus might mean for testing the Red Queen.

It seems to me that identification of the F locus is the most significant finding within this manuscript. However, the F locus is not specifically mentioned in the abstract (or the title). I find it quite interesting that what seemed a relatively straightforward host-pathogen interaction has seemingly become more and more complex with further investigation. This is noteworthy and would seem worth highlighting. This is merely a suggestion, but an introduction more focused on the evolution and genetic architecture of host resistance and a discussion more focused on the implications of finding this additional locus may allow the manuscript to stand on its own more effectively.

**Have all data underlying the figures and results presented in the manuscript been provided?**

Reviewer #1: Yes

Reviewer #2: Yes

PLOS authors have the option to publish the peer review history of their article (what does this mean?). If published, this will include your full peer review and any attached files.

Reviewer #1: No

Reviewer #2: No

---

## [Decision Letter · Decision Letter 1]

14 Dec 2022

Dear Dr Fredericksen,

We are pleased to inform you that your manuscript entitled "QTL study reveals candidate genes underlying host resistance in a Red Queen model system" has been editorially accepted for publication in PLOS Genetics. Congratulations! Please make sure you address the final reviewer comment about the stable GitHub link.

Yours sincerely,

Harmit S. Malik

Academic Editor

PLOS Genetics

Kirsten Bomblies

Section Editor

PLOS Genetics

Comments from the reviewers (if applicable):

Reviewer's Responses to Questions

**Comments to the Authors:**

Reviewer #1: The authors have addressed all my comments and the manuscript is greatly improved. I have one minor comment on data availability - the GitHub things should be given a stable doi (zenodo allows you to do this)

Reviewer #2: I believe the authors have sufficiently addressed my previous concerns regarding presentation of the work. The current version of the manuscript is an excellent contribution to the field. It clearly describes how the addition of the F locus provides a more complete picture of the host-parasite interaction. Further, I appreciate the discussion of the system in light of the F locus and its potential implications for coevolutionary dynamics. I believe this work is a significant advancement and an important piece to a complex, but worthwhile, puzzle.

**Have all data underlying the figures and results presented in the manuscript been provided?**

Reviewer #1: Yes

Reviewer #2: None

PLOS authors have the option to publish the peer review history of their article (what does this mean?). If published, this will include your full peer review and any attached files.

Reviewer #1: No

Reviewer #2: No

**Data Deposition**

http://datadryad.org/submit?journalID=pgenetics&manu=PGENETICS-D-22-00814R1

**Press Queries**

---

## [Editor Report · Acceptance letter]

9 Jan 2023

PGENETICS-D-22-00814R1 

QTL study reveals candidate genes underlying host resistance in a Red Queen model system 

Dear Dr Fredericksen, 

We are pleased to inform you that your manuscript entitled "QTL study reveals candidate genes underlying host resistance in a Red Queen model system" has been formally accepted for publication in PLOS Genetics! Your manuscript is now with our production department and you will be notified of the publication date in due course.

With kind regards,

Orsolya Voros

PLOS Genetics

On behalf of:
